# Production of Recombinant Gelonin Using an Automated Liquid Chromatography System

**DOI:** 10.3390/toxins12080519

**Published:** 2020-08-13

**Authors:** Maria E. B. Berstad, Lawrence H. Cheung, Anette Weyergang

**Affiliations:** 1Department of Radiation Biology, Institute for Cancer Research, Norwegian Radium Hospital, Oslo University Hospital, 0379 Oslo, Norway; maria.elisabeth.berstad@rr-research.no; 2Department of Experimental Therapeutics, The University of Texas MD Anderson Cancer Center, Houston, TX 77030, USA; lcheung@mdanderson.org

**Keywords:** recombinant, ribosome inactivating protein, gelonin, IMAC purification, protein expression, automated liquid chromatography, photochemical internalization

## Abstract

Advances in recombinant DNA technology have opened up new possibilities of exploiting toxic proteins for therapeutic purposes. Bringing forth these protein toxins from the bench to the bedside strongly depends on the availability of production methods that are reproducible, scalable and comply with good manufacturing practice (GMP). The type I ribosome-inhibiting protein, gelonin, has great potential as an anticancer drug, but is sequestrated in endosomes and lysosomes. This can be overcome by combination with photochemical internalization (PCI), a method for endosomal drug release. The combination of gelonin-based drugs and PCI represents a tumor-targeted therapy with high precision and efficiency. The aim of this study was to produce recombinant gelonin (rGel) at high purity and quantity using an automated liquid chromatography system. The expression and purification process was documented as highly efficient (4.4 mg gelonin per litre induced culture) and reproducible with minimal loss of target protein (~50% overall yield compared to after initial immobilized metal affinity chromatography (IMAC)). The endotoxin level of 0.05–0.09 EU/mg was compatible with current standards for parenteral drug administration. The automated system provided a consistent output with minimal human intervention and close monitoring of each purification step enabled optimization of both yield and purity of the product. rGel was shown to have equivalent biological activity and cytotoxicity, both with and without PCI-mediated delivery, as rGel_ref_ produced without an automated system. This study presents a highly refined and automated manufacturing procedure for recombinant gelonin at a quantity and quality sufficient for preclinical evaluation. The methods established in this report are in compliance with high quality standards and compose a solid platform for preclinical development of gelonin-based drugs.

## 1. Introduction

The development of protein toxins as cancer therapeutics is still in its infancy. Despite numerous reports on successful delivery of protein toxins from plants, unmodified or as part of fusion proteins, to cancer cells both in vitro and in vivo, their clinical potential remains largely unrealized [1]. Building the translational bridge from basic research to clinical drug application is a complex and multifaceted process. One of the main identified reasons for high failure rates and discontinuation of clinical drug development is the inability to reproduce preclinical data [2]. The methods used in small scale in the research lab are not necessarily translatable to industrial scale and biomedical companies invest both time and money in optimizing procedures that may not be replicable. A study published in 2015 estimates that, in the United States alone, approximately 28 billion USD are invested annually on preclinical research that is not reproducible [3]. To support clinical drug translation, there is a need for standardized and scalable methods that minimize sources of irreproducibility. System automation reduces human intervention and, consequently, the risk of human errors and provides a consistent output minimally affected by variations in time, production facilities or batch size. Minimizing the operator handling also helps maintaining safety requirements even at large scale. Establishing automated methods that fulfill good manufacturing practice (GMP) already at the basic research level should serve as an investment incentive for biomedical companies.

The type I ribosome-inactivating protein (RIP), gelonin derived from the seeds of *Gelonium multiflorum*, is an extremely potent inhibitor of protein synthesis [4]. Gelonin exerts *N*-glycosidase activity on the 28S ribosomal RNA unit of eukaryotic ribosomes by cleaving out adenine at the 4324 site, which irreversibly inhibits protein synthesis, resulting in cell death [5]. Analysis of the amino acid sequence of purified gelonin enabling the construction of a synthetic DNA encoding gelonin and production of a recombinant version of the gelonin toxin was achieved by two related groups in the mid 1990′s [6,7]. Since then, advances in recombinant DNA technology and better understanding of the structure and catalytic residues of gelonin [8] have made it possible to design and produce various gelonin fusion proteins that maintain the enzymatic activity of single gelonin [9,10,11]. It has been estimated that only a few gelonin molecules are needed to enter the cytosol for cell death to occur [12]. When introduced to living cells, gelonin has, however, limited cytotoxic potential due to entrapment and subsequent degradation in endosomes and lysosomes [13]. Several attempts have been made to circumvent challenges related to intracellular delivery of gelonin including various membrane-active and/or pore-forming peptides [11,14,15,16] and toxic gene therapy [17,18]. However, although these methods improve the cellular potency of gelonin considerably, subsequent toxicity to normal cells arises as a major hurdle which must be overcome for therapeutic applicability. Photochemical internalization (PCI) represents a minimally invasive strategy for endosomal drug release [19]. This method promotes photochemical rupture of endocytic vesicles by light-mediated activation of photosensitizers accumulated in endosomal and lysosomal membranes of tumor cells [20]. PCI releases the potential of gelonin only in illuminated areas and, hence, improves not only its efficiency, but also its selectivity. In addition, the immunogenicity of gelonin can be eliminated as PCI has been shown effective with only one single treatment [21]. The potentiation of gelonin toxicity by PCI was first proven in 1999 [19] and has since then been extensively studied both in vitro and in vivo [22,23,24,25]. The relatively low molecular weight (~30 kDa) of gelonin combined with the lack of a cell-binding domain enables engineering of fusion proteins at clinically relevant sizes that target and, upon endosomal release, induce specific cytotoxicity in tumor cells.

Our focus is on the development of gelonin-based drugs that to a large extent depend on PCI activation for efficacy, offering unique treatment selectivity [26]. To the best of our knowledge, all previous reports on production of gelonin and gelonin-based fusion proteins are based on manual chromatography techniques that may not be sustainable in industrial scale [27,28]. The aim of this study was to produce recombinant gelonin (rGel) according to the sequence published by Nolan et al. [7] at high purity and high yield using an automated liquid chromatography system. Close monitoring of the output of each chromatography step enabled optimization of the production procedure and the result was high quality gelonin with substantially less effort and use of resources than previously demonstrated. Endotoxin contaminations was minimized to a level compatible with parenteral drug administration [29]. Establishment of GMP-compliant facilities that supply gelonin and, further, gelonin-based drugs at high quality and quantity is of great interest as this will support future bench-to-bedside translation.

## 2. Results and Discussion

### 2.1. Cloning, Expression and Small-Scale Production of rGel

The DNA fragment of rGel was synthesized with BspHI and XhoI restriction sites introduced in the 5′ and 3′ end, respectively. The rGel insert was then cloned into the expression vector pET-32a using the XhoI and NcoI restriction sites (Figure 1a). Both the construction of the pET-32a/rGel expression vector and the subsequent DNA sequencing were done by Thermo Fisher GeneArt services. The amino acid sequence of rGel is provided in Figure 1b. Deletion of the NcoI site at nucleotide 108–113 by silent mutation did not alter the amino acid sequence (affected amino acids highlighted) in Figure 1b. 

The pET-32a/rGel plasmid was transformed into *E. coli* soluBL21, a strain optimized for the expression of soluble toxic proteins. To optimize the conditions for rGel expression, two different bacterial colonies transformed with PET-32a/rGel were cultured and induced by IPTG in three different media in small-scale (50 mL bacterial suspension). The bacterial growth was comparable for the two colonies within each culture media and greater in complete LB as compared to 50% LB and M9 media (results not shown). Also, the estimated amount of soluble protein at the expected molecular weight for His-tagged rGel (45.5 kDa) both before and after immobilized metal affinity chromatography (IMAC) was greater for LB. The estimated production yield after IMAC was ~7 mg per liter induced culture in complete LB, ~3 mg per liter induced culture in 50% LB and ~4 mg per liter induced culture in M9. The purity of gelonin after IMAC was similar for the different culture media (Figure 1c) and recombinant enterokinase (rEK) digestion resulted in products with identical size as evaluated by SDS-PAGE (data not shown). Based on this, complete LB medium was chosen as the most optimal culture condition for large-scale production.

### 2.2. Large-Scale Expression and Purification

The rGel production was further upscaled and the purification procedure was optimized to obtain rGel at high purity with respect to both protein contaminations and endotoxin level. The final purification procedure is depicted in Figure 2. The chromatogram profiles allowed close monitoring of each purification step and efficient selection of different fractions of washes and elutes for analysis by SDS-PAGE (Figure 3a). 9L induction of rGel in LB broth resulted, after initial IMAC, in ~80 mg of a 46 kDa protein assumed to be rGel including the His-tag (Figure 3b, left). The yield was in accordance with the yield of the small-scale production indicating reproducibility of the method upon upscaling. The demonstrated high yield using a scalable and monitorable method holds high promise for transferring the production method to GMP standards.

During IMAC, some bound impurities were partially or completely removed from the column by the 5 mM imidazole wash (W) prior to elution while other contaminants were eluted with the target protein at 150 mM imidazole (1B5–1C11) (Figure 3a, upper and Figure 3b, left). The relatively high density of contaminating proteins made it difficult to estimate the amount of 46 kDa protein in the IMAC start lysate. Thus, the yield in subsequent purification steps was reported relative to that following the initial IMAC.

IMAC fractions 1B5–1C11 (total volume 19 mL) were concentrated to 15 mL and subjected to buffer exchange chromatography (BEC) into 20 mM Tris-HCl (pH 7.6)/100 mM NaCl to facilitate protein digestion by enterokinase (Figure 3a, lower and Figure 3b, right). The delayed conductivity peak as compared to the UV absorbance reflects the successful retardation of salt molecules in the column resin. BEC fractions 2A6–2B4 were pooled (total volume 22 mL) and it was concluded that BEC did not cause any detectable loss of target protein. The automated BEC was shown as considerably less time-consuming and resource-intensive as compared to the traditional method for buffer exchange, dialysis. Overnight incubation with rEK cleaved off the 17 kDa His-tag resulting in a protein migrating at ~28 kDa under reducing conditions (Cut, Figure 3c), in agreement with the molecular mass of rGel. Evaluation of precipitations following rEK digestion indicated some loss of a 28 kDa target protein. However, for quantification purposes, it should be noted that the amount of sample loaded relative to the total quantity was 44-fold higher in the precipitate sample as compared to the soluble cut protein. In conclusion, the total yield following rEK digestion was estimated to 75% of that obtained after IMAC.

To separate rGel from the hexahistidine-tag, the rEK-digested product was loaded onto a column containing Cibacron Blue F3G-A with affinity for gelonin (Figure 4a) [30,31].

The 17 kDa His-tag was completely washed out with 100 mM NaCl (W) and bound rGel was successfully eluted at high purity with 2 M NaCl (1A6–9). The estimated product yield of Cibacron Blue chromatography was ~50% of that obtained after IMAC. Cibacron Blue also upconcentrated the product as shown by the narrow elution peak and the 65% volume reduction of the sample. Finally, rGel fractions 1A6–9 from two subsequent rounds of Cibacron Blue chromatography (total volume 8 mL) were concentrated to 4 mL and subjected to buffer exchange chromatography into 1×PBS (Figure 4b). rGel-containing fractions 1A9–1B8 were combined.

### 2.3. Removal of Endotoxins by Capto Adhere

Bacterial endotoxins initiate a host inflammatory response that may develop into severe sepsis. It is especially important to keep strict control of endotoxin contaminations in parenteral drugs as these bypass the protective barriers of the skin and intestinal wall. Gelonin-based drugs in combination with PCI are given as a single injection. The minimum endotoxin dose expected to cause fever in humans after one single administration is 5 endotoxin units (EU)/kg [32]. PCI of gelonin has previously been implemented with a maximum of 2.5 mg gelonin injected per kg mice [24,33]. In studies where gelonin has been used strictly as a non-targeted control for gelonin-based targeted toxins, the dosage has been considerably reduced [26,34]. In general, compared to mice, humans require smaller drug dose on weight basis [35]. Hence, endotoxin levels ≤1 EU/mg protein were considered acceptable purity for parenteral administration of the produced gelonin [29]. Capto adhere chromatography has previously been shown efficient for endotoxin reduction and was here run in flow through mode with contaminants adsorbing to the column (Figure 4c). Two cycles were required for sufficient endotoxin removal from the rGel solution. Comparing the start lysates of the first (SL1, 1 mL fractions 1A9–1B8 from BEC) and second (SL2, 2 mL fractions 1A8–1B4 from 1st cycle of Capto adhere) cycle, it may appear as if rGel was lost in this process. However, the less dense protein band at 28 kDa in SL2 merely reflects the dilution of the sample as the volume was increased by each cycle. The total volume of rGel-containing fractions after 2nd round of Capto adhere was 38 mL. Final calculation showed no loss of rGel through Capto adhere chromatography.

### 2.4. Qantification of Final Purified Product and Endotoxin Levels

The single band purified rGel sample was concentrated by centrifugal filter units and sterile filtered to a total volume of 11 mL (Figure 4c). The quality of the purified product was found sufficient as no clear contaminations were detected upon safe stain evaluation of 2 μg (Figure 4c) and 4 μg (result not shown). The final concentration was evaluated by three different methods; SimplyBlue gel staining, DC protein assay and Beer-Lamberts law, and finally estimated to 3.56 mg/mL (Figure 5a). The quantification results of SimplyBlue gel staining and DC protein assay were coherent (average of 3.7 and 3.4 mg/mL, respectively), whereas determination by Beer Lambert’s law led to a somewhat lower concentration (2.6 mg/mL). This was, however, also reflected in the quantification of previously produced rGel (rGel_ref_ estimated to 4.05 mg/mL as compared to original value 5.34 mg/mL, data not shown) and the concentration of our rGel product was therefore adjusted accordingly (to 3.43 mg/mL). Altogether, the calculated average of all six estimated values presented in Figure 5a was 3.56 mg/mL. The total yield of 39 mg (~50% overall yield compared to yield after IMAC) equals 4.4 mg final product/L induced culture, which is >4 times higher than previously reported (1 mg/L) [11,14]. A recent report by Ding et al. claims an even higher yield after expression of soluble recombinant gelonin in *E. coli* BL21 (DE3) (6.03 mg/L) [28]. Compared to the purification process used in the present report (Figure 2), the process used by Ding et al. was, however, considerably simplified as it merely involved Nickel-nitrilotriacetic (Ni-NTA) IMAC and was only demonstrated in small-scale (400 mL). Also, the report did not mention digestion of the hexahistidine tag or separation of free His-tag from the gelonin solution, which in our study induced a total of 50% loss of target protein as compared to after the initial IMAC. Furthermore, Ding et al. did not document on their endotoxin level and did not include any purification step to reduce the level of endotoxins, procedures likely to decrease the overall yield of the production.

Remaining endotoxins in the purified rGel product was quantified with the *Limulus* Amebocyte Lysate (LAL) assay with a sensitivity range of 0.1–1 EU/mL. As stated in Section 2.3, only endotoxin levels <1 UE/mg protein were considered applicable for in vivo purposes [32]. No linear correlation was found between estimated endotoxin level in undiluted sample (0.68 EU/mg) as compared to 1:5 and 1:100 dilution (0.05 and 0.08 EU/mg, respectively) (Figure 5b). This may be due to undiluted sample exceeding the range of linear correlation between absorption and endotoxin concentration. Hence, it was concluded that the endotoxin level of the rGel product was well below the set endotoxin limit concentration.

### 2.5. Biological Activity and Identity of rGel

The ability of rGel to inhibit translation was assessed in a cell-free reticulocyte lysate assay. rGel_ref_ produced previously by a non-automated process in the Rosenblum lab (described in the Materials and Methods Section 4.6) was used as control. The data points from four (rGel_ref_) and six (rGel) separate experiments, respectively, were plotted in the same graph page and subjected to sigmoidal regression (Figure 5c). The associated IC_50_ values of 104.1 pM (rGel_ref_) and 65.4 pM (rGel) appears consistent with previous reports on unmodified recombinant gelonin documenting IC_50_ values from 7.25 to 185 pM [6,9,11,26,36,37] and reveals that our rGel maintains equipotent protein synthesis inhibition activity.

The identity of rGel was examined by western blot using an anti-gelonin antibody and with rGel_ref_ as a positive control. As shown in Figure 5d, rGel and rGel_ref_ migrating at 28 kDa reacted with the antibody confirming the production of gelonin.

### 2.6. Cytotoxicity of rGel

The cytotoxicity of untargeted rGel among different cell lines may reflect differences in pinocytosis rate, intracellular trafficking, enzymatic activity and sensitivity to protein synthesis inhibition. The toxicity of rGel was evaluated in three cell lines for which data on recombinant gelonin efficacy is already reported with IC_50_ values ranging from nanomolar (MDA-MB-231 < 100 nM [38,39], A-431 < 500 nM [13,26]) to micromolar (CT26.WT < 5 µM) [11,14] concentration. The results on rGel cytotoxicity in the selected cell lines were in accordance with previous reports (Figure 6a, representative experiments). rGel_ref_ was used as a control with IC_50_ values shown comparable to that of the newly produced rGel.

The cellular potency of gelonin is severely hampered by lysosomal sequestration and degradation [13] and, hence, may be increased by PCI-induced cytosolic release [19]. PCI of rGel reduced the viability of CT26.WT cells both in a light dose-dependent (left) and toxin dose-dependent manner (right) as compared to rGel and photochemical treatment (photosensitizer + light) administered separately (Figure 6b). PCI at increasing rGel concentration was assessed using a light dose where the photochemical treatment itself reduced viability by 35–60%. The efficacy of PCI was significant at 10 nM rGel with no reduction in viability induced by rGel alone. PCI of rGel was equally effective as PCI of rGel_ref_ and the demonstrated activity of gelonin with PCI was in line with that previously reported in CT26.WT specifically [40] and in vitro in general [19,24,41].

## 3. Conclusions

This is, to the best of our knowledge, the first report on the cloning and soluble expression of rGel with subsequent purification using an automated liquid chromatography system. The expression and purification process was demonstrated as highly efficient, reproducible and predictable with minimal loss of target protein. Initial IMAC purification of 9 litre induced bacterial culture yielded ~80 mg rGel product. rEK digestion generated 60 mg cut target protein, i.e., 25% loss, and subsequent purification with Blue sepharose chromatography, BEC and Capto adhere chromatography yielded 39 mg final product (4.4 mg/L induced culture, ~50% overall yield compared to after IMAC), which is >4 times higher than previously reported (1 mg/L) [11]. The assessed endotoxin level of 0.05–0.09 EU/mg was well within the acceptable limit for parenteral drug administration. rGel was shown equally active and cytotoxic, both alone and with PCI-induced release, as rGel_ref_ produced using manual chromatography techniques. The automated methods established in this study were shown reproducible and scalable and facilitate further development and evaluation of rGel-based fusion toxins that, in combination with PCI, may target tumor cells with high specificity and efficiency.

## 4. Materials and Methods

### 4.1. Cell Lines and Culture Conditions

A-431 (CRL-1555), MDA-MB-231 (HTB-26) and CT26.WT (CRL-2638) cells were obtained from ATCC (Manassas, VA, USA) and maintained in DMEM (A-431) or RPMI-1640 medium (Sigma Aldrich, St Louis, MO, USA) supplied with 10% FCS, 100 U/mL penicillin and 100 μg/mL streptomycin. The cell lines were used between passage numbers 1–25 and routinely checked for *Mycoplasma sp.* infections.

### 4.2. pET-32a/rGel Plasmid Construction

The DNA fragment specifying the recombinant version of gelonin (rGel, 251 amino acids) was artificially synthesized according to *E. coli* codon preference and subcloned into the pET-32a vector (69015, Novagen, Merck KGaA, Darmstadt, Germany) using XhoI and BspHI/NcoI restriction sites. An internal NcoI site (nucleotide 108–113) was removed by silent mutation (CCA TGG → TCA CGG). Gene synthesis and subcloning, including sequence verification documenting the precise order of nucleotides, were provided by Thermo Fisher Scientific GeneArt services (Regensburg, Germany). The constructed pET-32a/rGel plasmid was handled according to the manufacturer’s instructions and transformed into *Escherichia coli* strain soluBL21 for protein expression.

### 4.3. Protein Expression

Bacterial colonies transformed with the pET-32a/rGel plasmid were first cultured in small-scale (50 mL) in complete LB broth, 50% LB or M9 medium and further in large-scale (9 L) in LB broth supplemented with 100 µg/mL ampicillin (LB_amp_) at 37 °C overnight in a shaker incubator at 200 rpm. The cultures were diluted 1:100 in fresh LB_amp_ and grown to A_600_ ~ 1.2 at 37 °C before another 1:1 dilution in LB_amp_. Thereafter, protein synthesis was induced at 23 °C overnight by addition of isopropyl β-d-thiogalactoside (IPTG) at a final concentration of 80 µM. The cells were collected by centrifugation at 9600× *g* for 10 min at 4 °C using a Sorvall™ RC 6 Plus Centrifuge and a Fiberlite™ F10-4 × 1000 LEX Fixed Angle Rotor (Thermo Fisher Scientific, Waltham, MA, USA).

### 4.4. Protein Purification

The bacterial cells were resuspended 1:20 in 40 mM Tris-HCl (pH 7.6)/300 mM NaCl and lysed using an LM20 Microfluidizer (Microfluidics, Westwood, MA, USA) at 4 °C with supplied pressure set to 1000 bar. The lysate was then ultracentrifuged at 120,000× *g* for 40 min at 4 °C using a Beckman Coulter Optima L-90K ultracentrifuge and a Type 45 Ti Fixed-Angle Titanium Rotor (Beckman Coulter, Brea, CA, USA).

All further purification steps were performed using the ÄKTA Avant system (GE Healthcare, North Richland Hills, TX, USA) with the fraction collector precooled to 6 °C and monitored by SDS-PAGE using 4–15% Mini-Protean TGX Precast Protein Gels (Bio-Rad Laboratories, Hercules, CA, USA), SimplyBlue SafeStain (Thermo Fisher) and Perfect Protein Markers, 10–225 kDa (#69079-3, Merck KgaA).

The supernatant fraction collected after ultracentrifugation was loaded onto a cobalt-containing HiTrap TALON Crude column (GE Healthcare), 1 mL column volume (CV) per 50 mL bacterial lysate) at a rate of 5 mL/min for IMAC. The column was washed with 5 CV 40 mM Tris-HCl (pH 7.6)/300 mM NaCl containing 5 mM imidazole and eluted with 8 CV 150 mM imidazole collected as 1 mL fractions in 96-well plates. Fractions containing His-tagged rGel (45 kDa), as evaluated on the chromatograms as well as on SimplyBlue-stained gels, were pooled and concentrated to a maximum volume of 15 mL using Amicon Ultra-15 Centrifugal Filter Units with a 10 kDa cut-off (Merck KgaA). The concentrated solution was then applied onto a HiPrep 26/10 Desalting column with Sephadex G-25 resin (GE Healthcare) at 10 mL/min for buffer exchange chromatography (BEC) into 20 mM Tris (pH 7.6)/100 mM NaCl (rEK buffer) collected as 2 mL fractions. rGel was digested overnight at room temperature under gentle agitation with rEK (P8070S, New England Biolabs, Ipswich, MA, USA, 0.00016 µg rEK per 25 µg of gelonin). The solution was centrifuged to remove precipitations following rEK digestion and the precipitation was re-suspended in rEK buffer, sonicated and spun down prior to SDS-PAGE analysis.

To remove free hexahistidine-tag from the product, the rEK-digested solution was passed through a 0.45 µm millipore membrane filter (Merck KgaA) and loaded at 1 mL/min onto a HiTrap Blue HP column (GE Healthcare), 1 mL CV per 6 mg cut protein) containing Cibacron Blue F3G-A with affinity for gelonin [30]. A maximum of 5 × 1 mL columns was run in sequence to avoid exceeding the set column pressure limit and, hence, the rGel product was applied in fractions according to the column capacity. The column was washed with 5 CV 20 mM Tris-HCl (pH 7.6)/100 mM NaCl and eluted with 8 CV 20 mM Tris-HCl (pH 7.6)/2 M NaCl collected as 1 mL fractions. Finally, the rGel solution was again concentrated and applied onto the HiPrep 26/10 Desalting column for BEC into 1 × DPBS (pH 7.5, Sigma) and collected as 1 mL fractions.

### 4.5. Endotoxin Removal and Detection

Removal of endotoxins was performed after flushing the ÄKTA Avant System with 1M NaOH followed by ddH_2_O. The purified rGel solution was loaded onto a HiTrap Capto Adhere 3 × 5 mL column (GE Healthcare) in flow through mode at 1 mL/min with the endotoxin contaminants adsorbing to the multimodal ion exchange resin. The flow through containing rGel was collected as 2 mL fractions. Endotoxin levels were estimated using Pierce LAL Chromogenic Endotoxin Quantitation Kit (#88282, Thermo Fisher) according to the manufacturer’s instructions. Endotoxin levels ≤1 EU/mg protein were considered acceptable purity for in vivo administration of the produced gelonin. Finally, the purified rGel solution was concentrated using Amicon Ultra-15 Centrifugal Filter Units with a 10 kDa cut-off, filter sterilized and stored at −80 °C in aliquots.

### 4.6. Verification of Concentration and Identity

Concentration of the purified sample was verified using three different methods. (1) SimplyBlue staining of proteins separated on SDS-PAGE gels and (2) DC Protein assay (Bio-Rad), performed according to the manufacturer’s microplate assay protocol, were both done using a previous version of recombinant gelonin (rGel_ref_) as a reference in addition to BSA as standards. rGel_ref_ has identical amino acid sequence as the rGel produced in the current study [7], but was produced in the Rosenblum lab using non-automated chromatography techniques [6]. In addition, Perfect Protein Markers, 10–225 kDa (Merck KgaA) were used as a reference for gel quantification. The gels were photographed with a Bio-Rad GS-800 scanner, and the software ImageLab 4.1 (Bio-Rad) was used for digital processing and quantification of protein bands. (3) The absorbance of the gelonin solution at 280 nm was used to estimate concentration according to Beer-Lambert’s Law:A = εm × C × l(1)
where A = absorbance, εm = molar extinction coefficient, C = concentration, l = path length of 1 cm [42].

The identity of rGel in the 28 kDa product was verified by western blot analysis. The purified sample was run on a 4–15% SDS-PAGE alongside Precision Plus Protein Dual Color Standards (#1610374, Bio-Rad) followed by transfer to a PVDF membrane using the Trans blot turbo system (Bio-Rad). After transfer, the membrane was blocked with 5% dry milk for one hour at room temperature, incubated with an in-house made rabbit α-gelonin polyclonal antibody (produced in the Rosenblum lab, 2 µg/mL in 5% dry milk) overnight at 4 °C, washed with TBST, and incubated with α-rabbit secondary antibody (1:2000, #7074, Cell Signaling Technology, Danvers, MA, USA) for one hour at room temperature. Finally, the membrane was washed with TBST, stained with SuperSignal West Dura Extended Duration Substrate (#34075, Thermo Fisher), acquired using ChemiDoc Gel Imaging System (Bio-Rad) and processed using the software ImageLab 4.1.

### 4.7. Protein Translation Inhbition Assay

The ribosome-inhibiting activity of rGel_ref_ or rGel was assessed in a cell-free system as previously described [43]. The data points from all experiments were combined and plotted in the same graph page and IC_50_ values were calculated from sigmoidal curves (fit model: a/(1 + exp(−(x − x0)/b))).

### 4.8. Cytotoxicity Assays

A-431, MDA-MB-231 and CT26.WT cells were seeded in 96-well plates at 3000, 5000 and 3000 cells/well, respectively, and allowed to attach overnight. The cells were then treated with rGel or rGel_ref_ at indicated concentrations for 72 h before cellular viability was assessed by MTT assay as previously described [44].

The cytotoxicity of rGel-PCI was assessed in CT26.WT cells. The cells were seeded in 96-well plates at 3000 cells/well and allowed to attach for 6 h before 18 h treatment with 0.4 μg/mL of the photosensitizer TPCS_2a_ (disulfonated tetraphenylchlorine, Amphinex) provided by PCI Biotech, Oslo, Norway [45]. The cells were then washed twice in PBS and incubated 4 h with rGel or rGel_ref_ before supplied with fresh medium and exposed to light from LumiSource (PCI Biotech) as indicated in the figure legends. Cell viability was assessed 48 h after light exposure by MTT assay [44].

## Figures and Tables

**Figure 1 toxins-12-00519-f001:**
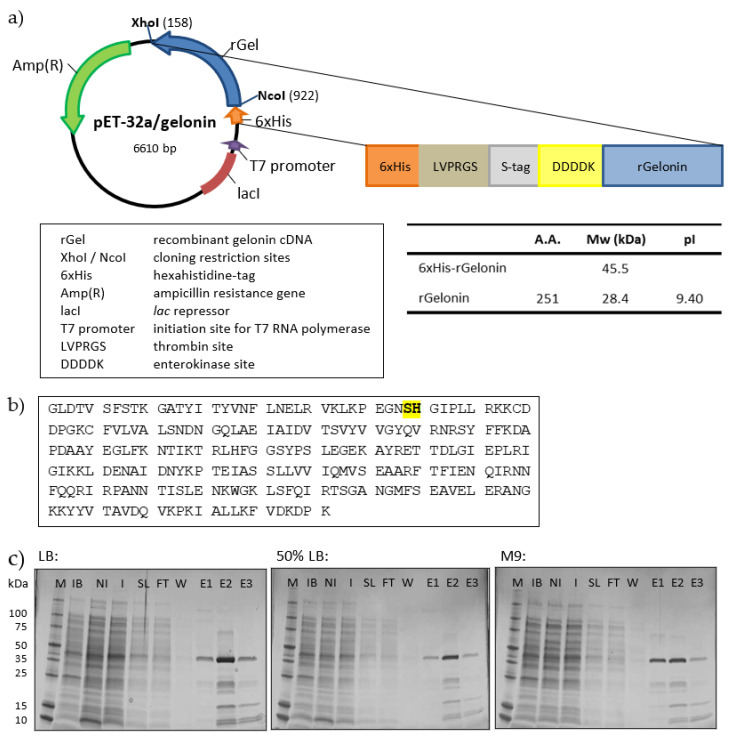
(**a**) Schematic representation of the pET-32a/rGel vector showing location of the different features. (**b**) Amino acid sequence of rGel. Site of silent mutation is highlighted. (**c**) SDS-PAGE under reducing conditions of IMAC-purified proteins expressed in LB, 50% LB or M9 culture media. Lane M, perfect protein marker (contains 0.5 μg for each of the protein sizes, except 1.0 μg for the 50-kDa band); IB, inclusion bodies; NI, non-induced culture; I, induced culture; SL, IMAC start lysate; FT, flow through; W, 5 mM imidazole wash; E1, E2 and E3, 150 mM imidazole eluted fractions.

**Figure 2 toxins-12-00519-f002:**
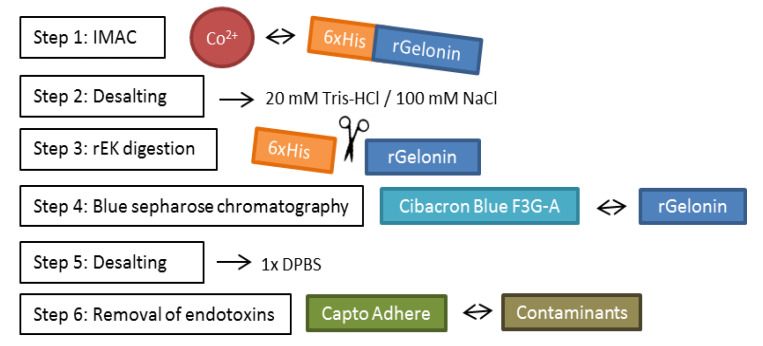
Schematic diagram of the rGel purification process.

**Figure 3 toxins-12-00519-f003:**
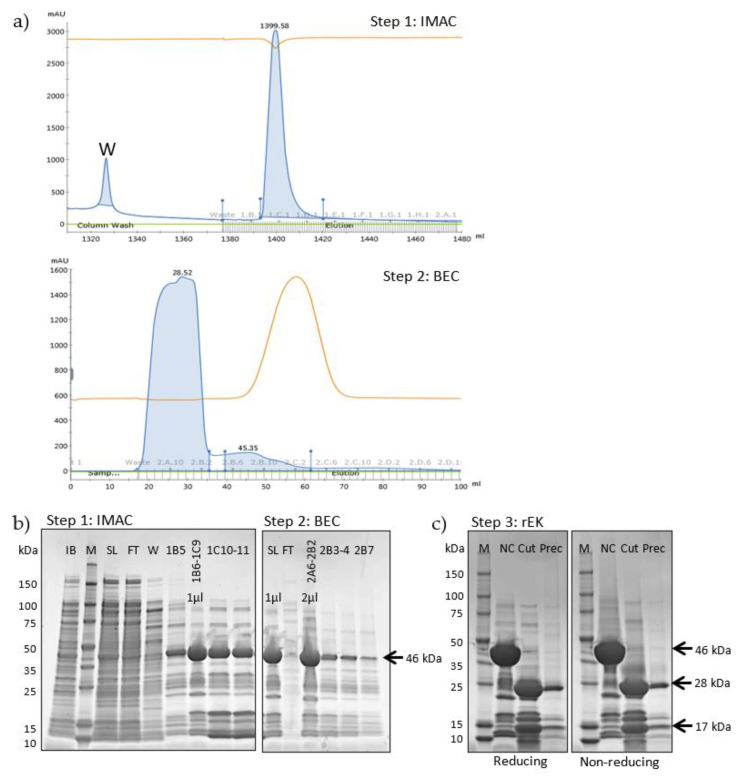
(**a**) Chromatogram of IMAC (upper) and BEC (lower) of 6×His-rGel. Vertical axis; UV absorbance (280 nm). Conductivity displayed in orange. Horisontal axis; eluent volume (lower) and collected fractions (upper). (**b**) SDS-PAGE under reducing conditions showing the yield and purity of 6×His-rGel following IMAC (left) and BEC (right). Lane M, perfect protein marker (contains 0.5 µg for each of the protein sizes, except 1.0 µg for the 50-kDa band); IB, inclusion bodies; SL, start lysate; FT, flow through; W, 5 mM imidazole wash; IB5–1C11, 150 mM imidazole eluted fractions; 2A6–2B7, desalted fractions. Sample application volume: 8 µL/well unless stated otherwise. (**c**) SDS-PAGE analysis under reducing (left) or non-reducing (right) conditions of cleavage products after incubation of 6×His-rGel with rEK. NC (non-cut) and cut, undigested and digested gelonin, respectively. Prec, precipitations. Sample application volume: 8 µL/well.

**Figure 4 toxins-12-00519-f004:**
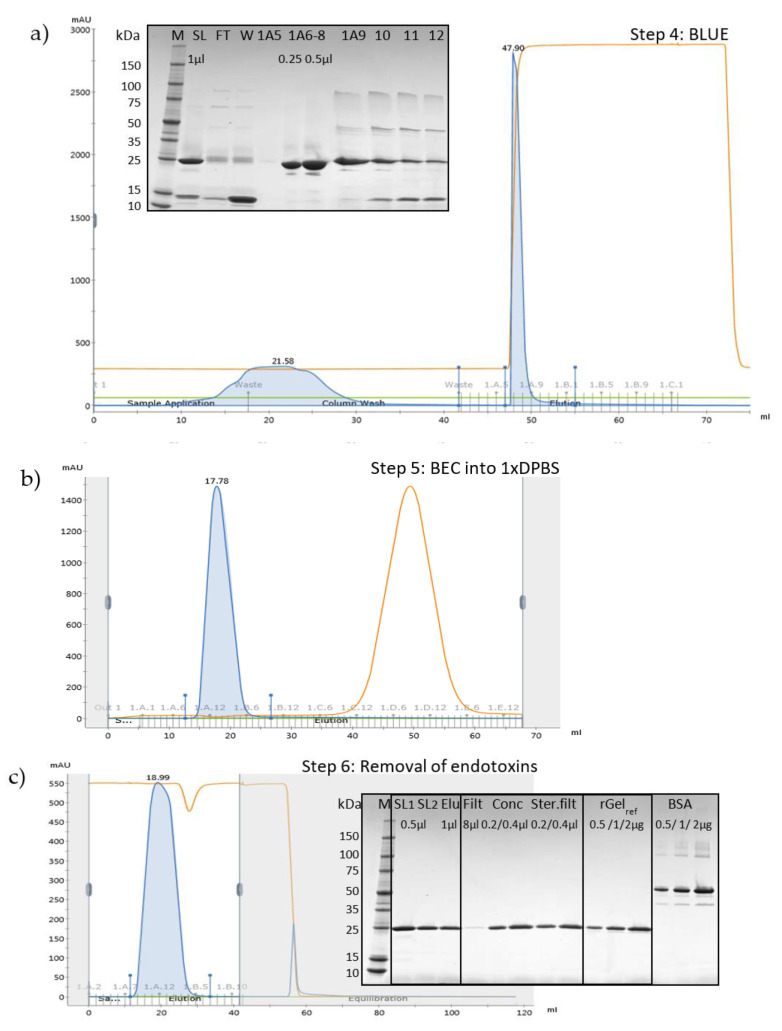
(**a**) Chromatogram and SDS-PAGE following removal of free hexahistidine tag by Blue sepharose affinity chromatography. Lane M, perfect protein marker (contains 0.5 µg for each of the protein sizes, except 1.0 µg for the 50-kDa band); SL, start lysate; FT, flow through; W, 100 mM NaCl wash; 1A5–1A12, 2 M NaCl eluted fractions. Sample application volume: 8 µL/well unless stated otherwise. (**b**) Chromatogram of BEC of rGel into 1×DPBS. (**c**) Chromatogram of 1st cycle of Capto adhere multimodal chromatography of rGel and SDS-PAGE following two subsequent rounds of Capto adhere. SL1/SL2, start lysate of 1st and 2nd round of capto adhere, respectively; Elu, eluent after 2nd round of capto adhere, Conc, concentrated rGel, Ster.filt, filter sterilized rGel; rGel_ref_ and BSA, standards. Gels in (**a**,**c**) run under reducing conditions. For chromatograms: UV absorbance at 280 nm shown on the vertical axis, eluent volume (lower) and collected fractions (upper) shown on the horisontal axis. Conductivity displayed in orange.

**Figure 5 toxins-12-00519-f005:**
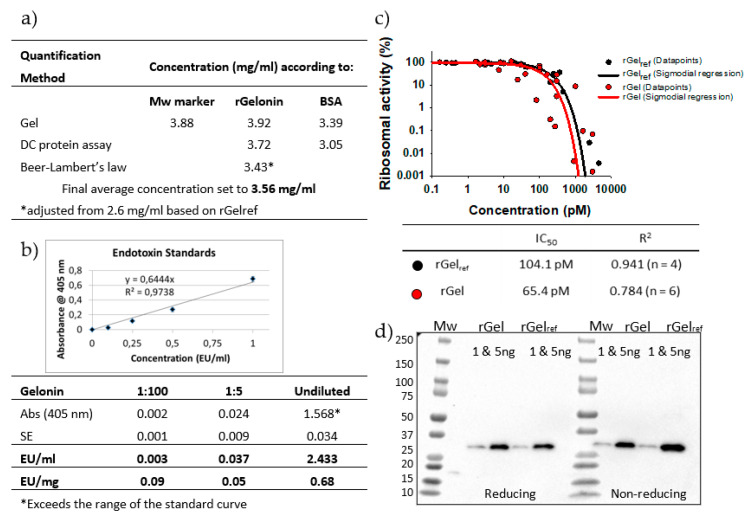
(**a**) Table displaying the concentration of purified rGel as quantified by SimplyBlue Gel stain, DC protein assay and Beer-Lambert’s law. The average of all six calculations was set as the final stock solution concentration. (**b**) Quantification of bacterial endotoxins in purified rGel as determined by LAL test. Endotoxin units per ml (EU/mL) or mg (EU/mg) of undiluted rGel (3.56 mg/mL) or 1:5 or 1:100 rGel dilution was calculated based on the endotoxin standard curve. (**c**) Relative ribosomal activity in reticulocyte lysate after incubation with rGel_ref_ or rGel. Sigmoidal regression of data points combined from a total of four (rGel_ref_) and six (rGel) experiments, respectively, and associated IC_50_ and R^2^ values. (**d**) Western blot analysis of 1 or 5 ng rGel_ref_ or rGel under reducing and non-reducing conditions using an anti-gelonin antibody.

**Figure 6 toxins-12-00519-f006:**
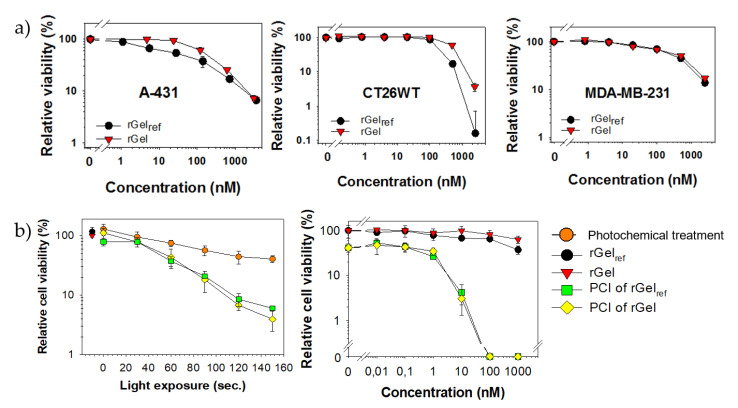
(**a**) Cellular viability (MTT) of the respective cell lines after 72 h incubation with increasing concentrations of rGel_ref_ or rGel. (**b**) PCI of rGel_ref_ or rGel in CT26.WT cells. Light dose-dependent PCI of 10 nM rGel_ref_ or rGel (left panel) and toxin dose-dependent PCI applying 60 s of light (right panel). The graphs presented are representative of three separate experiments. Results are the average of triplicates normalized to non-treated cells. Error bars = s.d.

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
