# Peer review of "Production of Recombinant Gelonin Using an Automated Liquid Chromatography System"

_toxins, 2020, doi:10.3390/toxins12080519_

Round 1

Reviewer 1 Report

The manuscript “Production of recombinant gelonin using an automated liquid chromatography system” describes the automated purification of recombinant gelonin. Authors showed that the purified product had acceptable endotoxin and gelonin activity levels.

The manuscript is well written, and data well presented.

Minor comments:

Lines 16, 19 and 31. Please define your acronyms before use, etc. IMAC, rGel, BEC, etc.

Author Response

Reviewer 1 comment:

Lines 16, 19 and 31. Please define your acronyms before use, etc. IMAC, rGel, BEC, etc.

Response:

Thank you pointing out this deficiency. All acronyms are now defined at their first appearance. (The original draft was written with Materials and Methods in front of the Results and Discussion section. Hence, the acronyms in our previous submission was defined in the M&M).

Reviewer 2 Report

Gelonin is a plant protein, a toxin, that shows high potential in cancer treatment, as it inactivates ribosomes. Therefore, there has been a demand for getting higher amounts of a soluble and pure protein for further studies as well as therapeutic applications.

While the paper is well written and all procedures clearly described, I am not sure what is so new here in comparison to the recent publication of Ding et al (in the list of references in the manuscript under No. 28). It is true, that in Ding et al., the construct still contained His tag, but this could be easily improved by putting in the TEV or other specific protease site to remove it. This would probably be a much shorter and probably more efficient procedure that the one described in this reviewed manuscript. Also, on the gel on Fig 4C, the amount of the loaded rGel is low, so how would the gel look if more of the protein was loaded. Would it still look as clean as now?

In addition, I do not understand why the authors expose the fact that they used an automated liquid chromatography system. AKTAs (and similar systems) have been quite standard.

Also: are references 12 and 21 appropriately addressed?

Author Response

Reviewer 2 comment 1:

While the paper is well written and all procedures clearly described, I am not sure what is so new here in comparison to the recent publication of Ding et al (in the list of references in the manuscript under No. 28). It is true, that in Ding et al., the construct still contained His tag, but this could be easily improved by putting in the TEV or other specific protease site to remove it. This would probably be a much shorter and probably more efficient procedure that the one described in this reviewed manuscript.

Response 1:

Here we do not agree with the reviewer. The product generated by Ding et al. has only been purified by IMAC and therefore still contains the His-Tag used for this purification. Also, Ding et al. have not documented their endotoxin level, which probably is high since they have not included any procedures for endotoxin removal in their production. The presence of the His-Tag as well as the non-documented level of endotoxins is why the product generated by Ding et al. is less suited for preclinical evaluation (the drug authorities require no artificial tags and an endotoxin level below 1 UE/mg protein for clinical evaluation). If Ding et al. were to include purification steps involving His-tag cleavage (with e.g TEV protease or similar) and subsequent His-tag removal in addition to endotoxin reduction, this would clearly result in a more comprehensive purification procedure and, most likely, reduce the gelonin yield. As an example, cleaving off the His-tag With TEV protease or similar often causes precipitation as well as digestion of the protein product.  In the new version of our manuscript, we have complemented our discussion on the yield of Ding et al. in paragraph 2.4 page 7 to clarify these important points.

We also think it is important to note that Ding et al. used a manual production procedure involving only 400 ml induced bacteria with no documentation of the scalability or reproducibility of the methods used. Development of gelonin-based drugs for clinical purposes requires scalable methods that comply with GMP. The automated procedure established in the current study was developed with this in mind and was shown scalable and monitorable from 50 ml to up to 9 L induced bacteria per batch. In our revised version of our manuscript we have addressed this point more clearly in paragraph 2.2 at page 4.

Reviewer comment 2:

Also, on the gel on Fig 4C, the amount of the loaded rGel is low, so how would the gel look if more of the protein was loaded. Would it still look as clean as now?

Response 2:

In the gel in Fig 4c, the largest band of sterile filtered rGel contains 1.4 μg, whereas SL1 contains 2 μg. The apparent low amount of impurities with this loading implies a product with low amount of purities. We have also evaluated 4 μg rGel with no impurities identified. These points are now included in the manuscript at page 7. We also believe it is worth pointing out that the largest band of rGelref and BSA in Fig. 4c contains 2 μg each with minor impurities clearly visible for both of these proteins.

Reviewer comment 3:

In addition, I do not understand why the authors expose the fact that they used an automated liquid chromatography system. AKTAs (and similar systems) have been quite standard.

Response 3:

Automated Liquid chromatography systems have indeed emerged as standards for recombinant production of other substances. However, to the best of our knowledge, this is the first study demonstrating automated purification of gelonin or gelonin-based drugs. The preclinical development of proteins for medicinal purposes relies on production methods that are traceable, reproducible and scalable complying with all requirements of GMP. The described methods for gelonin production using an automated system comply with these requirements and are therefore emphazised in this context. Please also find our answer to comment 1.

Reviewer comment 4:

Also: are references 12 and 21 appropriately addressed?

Response 4:

We believe these References are appropriately addressed. Ref 12 indicates that as little as 1-10 RIP molecules in the cytosol are sufficient to kill one cell. Ref 21 documents results of the first clinical trial of PCI of bleomycin.

Round 2

Reviewer 2 Report

No further comment